# Metaviromes Reveal the Dynamics of *Pseudomonas* Host-Specific Phages Cultured and Uncultured by Plaque Assay

**DOI:** 10.3390/v13060959

**Published:** 2021-05-21

**Authors:** Katrine Wacenius Skov Alanin, Laura Milena Forero Junco, Jacob Bruun Jørgensen, Tue Kjærgaard Nielsen, Morten Arendt Rasmussen, Witold Kot, Lars Hestbjerg Hansen

**Affiliations:** 1Department of Plant and Environmental Sciences, University of Copenhagen, 1871 Frederiksberg C, Denmark; Ksala@envs.au.dk (K.W.S.A.); laura.junco@plen.ku.dk (L.M.F.J.); jacobbruun@protonmail.com (J.B.J.); tkn@plen.ku.dk (T.K.N.); 2Department of Environmental Science, Aarhus University, 4000 Roskilde, Denmark; 3Department of Food Science, University of Copenhagen, 1958 Frederiksberg C, Denmark; mortenr@food.ku.dk; 4COPSAC, Copenhagen Prospective Studies on Asthma in Childhood, Herlev and Gentofte Hospital, University of Copenhagen, 2820 Gentofte, Denmark

**Keywords:** bacteriophages, viromes, metaviromes, Oxford Nanopore Technologies, Illumina sequencing, plaque-assay, PEG precipitation, phage enrichment, *Pseudomonas syringae* pv. *tomato* DC3000

## Abstract

Isolating single phages using plaque assays is a laborious and time-consuming process. Whether single isolated phages are the most lyse-effective, the most abundant in viromes, or those with the highest ability to make plaques in solid media is not well known. With the increasing accessibility of high-throughput sequencing, metaviromics is often used to describe viruses in environmental samples. By extracting and sequencing metaviromes from organic waste with and without exposure to a host-of-interest, we show a host-related phage community’s shift, as well as identify the most enriched phages. Moreover, we isolated plaque-forming single phages using the same virome–host matrix to observe how enrichments in liquid media correspond to the metaviromic data. In this study, we observed a significant shift (*p* = 0.015) of the 47 identified putative *Pseudomonas* phages with a minimum twofold change above zero in read abundance when adding a *Pseudomonas* *syringae* DC3000 host. Surprisingly, it appears that only two out of five plaque-forming phages from the same organic waste sample, targeting the *Pseudomonas* strain, were highly abundant in the metavirome, while the other three were almost absent despite host exposure. Lastly, our sequencing results highlight how long reads from Oxford Nanopore elevates the assembly quality of metaviromes, compared to short reads alone.

## 1. Introduction

As bacteriophages (phages) are the most abundant entity across all environments, they serve as a major reservoir of genetic diversity and have an essential role in ecology and maintaining bacterial diversity as well as carbon and nutrient cycling [1,2,3,4]. Phages keep dominant bacteria in check by ‘killing the winner’ through lysis, which results in the release of organic matter from the exterminated host used by other prokaryotes [5]. The ability of phages to attack and lyse very specific host bacteria has sparked a significant interest to discover novel phages to fight pathogens in phage therapy (human infections) and phage biocontrol (agriculture) [6,7,8,9,10,11].

It is estimated that the number of phages is at least ten-fold larger than the number of bacteria in the sea and that the number of phage particles on Earth is 10^31^ [4,12,13,14]. The combined biomass of phages on Earth is assumed to correspond to 75 million blue whales [3,15,16]. Despite their importance and high number, phages remain understudied, although research in phage genomics and viromics is on the rise. The number of available phage genomes in databases is rapidly increasing, especially uncultivated viral genomes (UViGs), but there is a lack of high-quality phage genome representatives in the databases [14].

One reason for the lack of high-quality phage genomes is that the methodology for isolating single phage strains with known hosts is laborious, time-consuming, and often the limiting step [17,18,19]. One of two methods, (I) direct plating or (II) enrichment approach, is most often used to isolates single phages. Enrichments are often used to increase the number of phages against the host of interest, and they work by incubating the host together with a phage-containing sample in growth media. Both ways are typically followed by the double-agar overlay technique first described by Gratia to observe plaque-forming units [20,21]. An additional step used in some cases is polyethylene glycol (PEG) precipitation, which concentrates and purifies the phages in a sample before direct plating or after enrichment [22,23]. Plaque assays are usually the final step of isolating phage–host pairs. It is a simple technique, but parameters such as the nature of the phage itself, insufficient phage titer, temperature, percentage of the agar in the overlay, presence of specific ions, or adsorption of phage particles to host cells can all affect whether or not a visible plaque will form [18,24]. Some phages are not capable of making plaques on plates due to limited diffusion in agar in general or low productivity, and isolation of highly effective single phages might be unsuccessful despite them being present and able to lyse the host [15]. All these restrictions limit the types and numbers of phages isolated, despite attempts to make it more high-throughput [17].

Recent advances in sequencing have enabled genomic investigation of phages from environmental samples through targeted sequencing of metaviromes (referred to as viromes in this study) [25]. This allows for investigating both plaque-forming and nonplaque-forming phages, giving a broader understanding of the type of phages in different environments and expanding phage genome databases.

Identification of nonplaque-forming phages targeting specific bacteria could improve the types of phages isolated against human, animal, or plant pathogens. Phages isolated by plaque assays may not be the most effective or the most abundant in the phage community but simply the best ones to make detectable plaques. Thus, using a virome from an environmental sample and extracting it with and without cultivation with a specific host can highlight what members of the viral community amplify in the presence of the host in a complex viral community. One of the most important characteristics of a phage is related to which hosts it lyses or interacts with. However, these characteristics are hard to predict on the basis of metavirome sequencing alone. Comparing the two types of viromes (cultivated with and without a specific host) to phages isolated from plaque assays using the same host–virome matrix will further help to predict phage–host interactions and indicate if the most abundant phages are plaque-forming or whether other (perhaps nonplaque-forming) phages are enriched.

This study uses organic waste (OW) samples together with the tomato pathogen *Pseudomonas syringae* pv. *tomato* DC3000 (DC3000) to analyze the community shift in the OW virome when exposed to the tomato pathogen. We compare the OW virome to five plaque-forming phages isolated by traditional plaque assay from the same virome–host matrix and demonstrate that only two of the plaque-forming phages are potentially among the most enriched in the virome. The setup of the experiment and key findings are depicted in Appendix A. This study raises important questions on how future phage isolation is to be done, how we can predict phage–host matrices from metaviromes, and how we can demonstrate the importance of long-read sequencing data when assembling viromes.

## 2. Materials and Methods

### 2.1. Bacterial Host Strain and Organic Waste Sample Preparation

*Pseudomonas syringae* pv. *tomato* DC3000 (DC3000) [26,27] was maintained in Lysogeny broth (LB) solid or liquid media [28] and incubated at room temperature (approximately 21 °C and shaking at 200 rpm). One of the largest waste-management company of Copenhagen (HCS A/S, Glostrup, Denmark) provided the liquid generated from common household organic waste (OW). It was kept at −20 °C after sampling to stop all biological activity until thawing and usage. To prepare the OW phage community for spiking/enrichment with DC3000 (explained in Section 2.3), 300 mL of OW sample was centrifuged for 10 min at 10,000× *g*, and the supernatant was filtered through PDVF syringe filters (0.45 μm, Merck Millipore, Darmstadt, Germany) to deplete organic and bacterial matters from the sample. The filtered OW sample was kept at 4 °C overnight.

### 2.2. Plaque Assay and Isolation of Single Phages

The five single phages (DrKristoffer, OtownIsak, SummerboyErik, GhostToast, and Hovsa) from the OW sample (HCS A/S, Glostrup, Denmark) were isolated using direct plating on 0.6% soft agar double overlay, the same way as the three phages described in Jørgensen et al. (2020) but using *P. syringae* pv. *tomato* DC3000 as the host [11,29]. For DrKristoffer and OtownIsak, the OW sample was PEG purified as described in Section 2.3 before direct plating. SummerBoyErik was isolated from enriched OW samples; aliquots of the OW sample were enriched overnight with *P. syringae* DC3000 before direct plating as described previously [11,29,30]. GhostToast and Hovsa were both isolated with direct plating from the original OW sample. The phage DNA extraction was made as described by Jørgensen et al. (2020) and Carstens et al. (2019) [11,29]. Sequencing libraries for DrKristoffer, GhostToast, OtownIsak, and SummerBoyErik were prepared using NEBNext^®^ Ultra II FS library prep kit for Illumina (New England Biolabs, Ipswich, MA, USA), while the sequencing library for Hovsa was prepared with Nextera^®^ XT DNA Kit (Illumina, San Diego, CA, US). The paired-end libraries were sequenced on the Illumina iSeq 100 platform (2 × 150 cycles). Adapter trimming and genome assembly were performed with CLC Genomics Workbench v. 12.0.03 with default settings.

### 2.3. Virome Spike and Extraction

The series of in-the-bottle infections, referred to as spiked viromes, was carried out in 200 mL of LB media to which 4 mL of the filtered OW virome sample was added together with MgCl_2_ and CaCl_2_ to a final concentration of 10 mM, and 1 mL of the overnight DC3000 or sterile water was added in the case of the non-spiked virome. Samples were incubated at room temperature with 200 rpm shaking for 8 h. The spiked virome and the non-spiked virome samples had technical triplicates. A modified PEG precipitation of the protocol ‘Harvesting, Concentration, and Dialysis of Phage’ (www.phagesdb.org last accessed 6 September 2019) was used for the initial concentration of the spiked virome and non-spiked virome, as well as 250 mL of filtered OW sample, from here on referred to as the baseline virome. The viromes with a concentration of 10% polyethylene glycol 8000 (Millipore, 1546605) and 1 M NaCl were left shaking at 200 rpm and 4 °C overnight, followed by centrifugation for 10 min at 10,000× *g* and 4 °C to pellet the PEG precipitate containing the phages and to remove the supernatant. The PEG pellets were resuspended in 12 mL of SM buffer (100 mM NaCl, 8 mM MgSO_4_, 50 mM Tris-Cl) each and relocated in 50 mL falcon tubes. To ensure complete homogenization of the PEG pellet, the tubes were left shaking at 200 rpm and 4 °C overnight. The samples were centrifuged at 4 °C and 10,000× *g* on the next day to remove PEG and recover the supernatant now containing the phages. The supernatants were filtered through 0.45 μm PVDF filters (Merck Millipore, Darmstadt, Germany) for optimal removal of PEG and potential bacteria. All viromes were concentrated from 12 mL down to approximately 200 μL of SM-buffer using the Amicon^®^ Ultra-15 Centrifugal Filter Devices 100K (Merck Millipore, Darmstadt, Germany), following the manufacturer’s recommendations. Both the spiked and non-spiked viromes were in technical triplicates, and the baseline virome had no replicates. Seven viromes were prepared and sequenced in this study.

### 2.4. DNA Extraction, Purification, and Library Prep

DNA extraction was done by the phenol-chloroform method [31]. Briefly, the phage samples were treated with 5 U of DNase I (A&A Biotechnology, Gdynia, Poland) for 30 min at 37 °C, followed by treatment with 0.5% (*v*/*v*) sodium dodecyl sulfate (SDS) and 6 U of proteinase K (A&A Biotechnology, Gdynia, Poland) for 1 h at 55 °C. The samples had 200 μL of 3 M NH_3_Ac added and were then treated with an equal amount of phenol–chloroform to remove proteins and bacterial debris. The aqueous phase was transferred to a new tube and the DNA was precipitated standard EtOH precipitation [31]. The DNA pellet was dissolved in sterile MilliQ water. DNA concentration and purity were measured with Qubit 2.0 fluorometer (Life Technologies, Carlsbad, CA, USA) and NanoDrop spectrophotometer (Thermo Scientific, Waltham, MA, USA), respectively. Libraries from all seven viromes were built according to the manufacturer’s instructions using the Nextera^®^ XT DNA Kit (Illumina, San Diego, CA, USA) and sequenced as paired-end reads on an Illumina NextSeq500 platform using the Mid Output Kit v2 (2 × 150 cycles). A barcoded Nanopore library was built for the baseline virome using Rapid Barcoding Sequencing kit (SQK-RBK004) (Oxford Nanopore Technologies, Oxford, UK) and sequenced on the MinION platform using an R9.4 flow cell (Oxford Nanopore Technologies, Oxford, UK). The Ligation Sequencing kit (SQK-LSK109) (Oxford Nanopore Technologies, Oxford, UK) was omitted to ensure the inclusion of potential circular phage genomes, which cannot have adaptors ligated. Sequencing and base-calling were done using MinKNOW v3.4.5 and Guppy v3.2.6, respectively. The purity and the DNA concentration are shown in Appendix A, together with the number of reads after quality control.

### 2.5. Quality Control and Assembly

The Mosaic workflow (https://github.com/lauramilena3/Mosaic, last accessed 8 March 2021) was used for quality control, assembly, identification of putative viral contigs, viral population clustering, and abundance table calculation.

In total, 12 assemblies were done: (1) baseline (Illumina only), (2) merged triplicates for DC3000 + (Illumina only), (3) DC3000 + A (Illumina only), (4) DC3000 + B (Illumina only), (5) DC3000 + C (Illumina only), (6) merged triplicates for DC3000 − (Illumina only), (7) DC3000 − A (Illumina only), (8) DC3000 − B (Illumina only), (9) DC3000 − C (Illumina only), (10) merged samples for baseline, DC3000 + triplicates, and DC3000 − triplicates, assembled with Illumina and scaffolded with baseline Nanopore reads, (11) baseline long-read assembly polished with Illumina merged samples for baseline, DC3000 + triplicates and DC3000 − triplicates, (12) baseline Nanopore assembly polished with baseline Illumina reads. Assemblies 1–9 used metaSPAdes, assembly 10 used hybrid metaSPAdes, and assemblies 11–12 used Canu (11 listed in Figure 1; 12 listed Appendix A).

For the quality control of Illumina reads, Trimmomatic v0.39 [32] was used to remove low-quality bases and adapters (LEADING:20 TRAILING:20 SLIDINGWINDOW:4:20 MINLEN:50). Trimmed reads were mapped with Kraken v2.1.1 against the minikraken2_v2_8GB_201904 database to detect bacterial contaminants. On the basis of bacterial contaminants, reads mapping to ΦX174 (NC_001422.1), *P. syringae* pv. *tomato* str. DC3000 (NC_004578.1), plasmid pDC3000A (NC_004633.1), plasmid pDC3000B (NC_004632.1), *Lacticaseibacillus paracasei* (NC_014334.2), and plasmid plca36 (NC_011352.1) were removed with BBDuk from BBTools suite v38.86 [33]. Reads mapping to *Lactobacillus* were still present, ranging from 10–11% in each of the seven viromes. Due to the lack of knowledge on the exact strain of *Lactobacillus* present in the OW, it was not possible to remove all *Lactobacillus* reads. Reads were normalized to ~100× target coverage and were error-corrected with bbnorm. For Nanopore reads, adapters were trimmed using Porechop v0.2.3 (https://github.com/rrwick/Porechop last accessed 16 March 2019), and low-quality bases were removed using NanoFilt v2.7.1 [34]. Long reads that mapped with minimap2 [35] to *P. syringae* or *L. paracasei* genomes were removed using SAMtools v1.10 [36].

For assemblies 1–9, downsampled reads were assembled with metaSPAdes v3.14.0 [37] with no error correction. For assembly 9, downsampled reads from each triplicate and trimmed Nanopore reads were used to perform a hybrid assembly with metaSPAdes. Assembly 10 of long reads was done with Canu v2.0 (genomeSize = 5 m minReadLength = 1000 corOutCoverage = 10000 corMhapSensitivity = high corMinCoverage = 0 redMemory = 32 oeaMemory = 32 batMemory = 200) [38]. This assembly was error-corrected with long reads using Racon v1.4.13 [39], and then polished with all the short reads combined using Pilon v1.23 [40]. The resulting contigs from the 11 assemblies were concatenated and filtered to a minimum contig size of 1000 bp and 2× coverage. Contigs denoted NODE_X are from metaSPAdes assemblies, and contigs denoted tig0000X are from the Canu assembly.

### 2.6. Viral Identification and Clustering

VirSorter v2.0 [41] was run on all the assembled contigs to detect putative viral sequences. Identified viral sequences were merged with genomes of the five plaque-isolated phages and then clustered into viral populations referred to as viral operational taxonomical units (vOTUs, presumably clustered at species level) using the stampede-clustergenomes script (https://bitbucket.org/MAVERICLab/stampede-clustergenomes/ last accessed 11 March 2019). Contigs were clustered at 95% nucleotide identity across 80% of the genome. The longest member of each cluster was selected as the representative. Completeness of representative contigs was assessed using CheckV v0.6.0 [42]. Contigs predicted as complete quality were within the high-quality group. Viral population representatives were mapped to NCBI RefSeq bacterial genomes (downloaded on 23 February 2021) using BLAST (v2.9.0+). Viral populations matching more than 40% of their length with a bacterial RefSeq hit were removed. Lastly, vOTUs that were predicted with CheckV as high-quality or medium-quality and contigs less than 5000 bp in those two groups were also selected for abundance analysis.

### 2.7. Abundance Calculations

Quality controlled reads for the spiked (DC3000 +) and non-spiked (DC3000 −) replicates were subsampled to the lowest number of reads (2,228,236). Subsampled and total reads were mapped against the filtered viral clusters using BBMap 2.3.5 (https://sourceforge.net/projects/bbtools/ last accessed 11 March 2019). SAMtools v1.10 [36] was used to generate a sorted bam. Contig depth coverage was calculated using tpmean from BamM 1.7.3 (https://github.com/Ecogenomics/BamM 11 March 2019). In the same way, contig breadth coverage was calculated with genomecov from BEDtools v2.29.2.0 [43]. For a contig to be considered present, more than 80% of the sequence needed to be covered. When this condition was not met, the abundance of the contig was set to zero. Viral population abundance tables for the total and subsampled reads were obtained, where the depth of coverage was normalized per million reads per sample. The relative abundance in each sample was calculated by dividing the contig depth coverage by the sum of all contig depth coverages in that sample.

### 2.8. Host Prediction

SpacePHARER v4 [44] was used to predict phage–host relationships at the genus level, on the basis of the detection of vOTUs that matched 363,460 CRISPR spacers from the Shmakov spacer dataset [45]. Alternatively, tblastx was used to align the vOTUs against a database of *Pseudomonas* phages. This database was obtained by downloading all 1253 entries in the nucleotide archive of NCBI, which had “*Pseudomonas* phage” in their description. These entries were filtered, and only 510 records that contained “complete genome” or “genome assembly” in their description were kept.

Contigs that matched a CRISPR spacer from a *Pseudomonas* host (SpacePHARER method) or contigs where more than 60% of the contig was covered by a sequence from the *Pseudomonas* phage database (tblastx method) were taken for manual validation. Those contigs were blasted using megablast, discontiguous megablast, or tblastx manually to ensure validity [46]. To deem a contig to be a putative *Pseudomonas* phage, the top viral hit, with the lowest E-value and highest query coverage percentage, had to be a *Pseudomonas* phage.

### 2.9. Diversity Analysis and Statistics

Conda package scikit-bio v0.5.5 (http://scikit-bio.org last accessed 11 March 2019) was used for diversity analysis.

Bray–Curtis dissimilarities between each sample were calculated, and the resulting distance matrix was used for ordination using principal coordinates analysis (PCoA) and hierarchical clustering of the samples. Furthermore, analysis of similarities (ANOSIM) was used to test whether our distances within and between category groups were different.

Differential abundance (DA) was done on the subsampled vOTU table using DAtest 2.7.17 [47], which, through simulations, screens a variety of DA tests for the ability to control both false negatives and false positives. The MetagenomeSeq feature model [48] was selected and used to analyze the differential abundance of vOTUs as a function of the sample being incubated with or without DC3000 (+/−). The log_2_ fold change (Log_2_FC) and false discovery rate (FDR)-adjusted *p*-values were calculated by the MetagenomeSeq feature model, indicating the differential read abundance between the two types of viromes for all the included contigs.

To reveal if there was an enrichment of *Pseudomonas* phages in the spiked virome, a Fisher exact test was conducted on the DA results, comparing ratios of the curated putative *Pseudomonas* phages to the putative *Lactobacillococcus* phages detected with the CRISPR method. *Lactobacillus* phages were selected as the reference due to their relatively high coverage and since they are naturally present in a high abundance in the OW sample.

## 3. Results

### 3.1. Viral Contigs Constitute the Majority of Each Assembly

A total of 12 assemblies were generated using either metaSPAdes (v3.14.0, assemblies 1–9), hybrid metaSPAdes (v3.14.0, assembly 10), or Canu (v2.0, assemblies 11–12) with the Illumina reads, Illumina and Nanopore reads, or Nanopore reads polished with Illumina reads, respectively. These 12 assemblies were composed of the baseline (assembly 1), one assembly for each of the DC3000 + A–C viromes (assemblies 2–4), as well as one assembly for the combined Illumina reads of DC3000 + (assembly 5). The last six assemblies were each of the DC3000 − A–C viromes (assembly 6–8), one of the combined reads Illumina of DC3000 − (assembly 9), one with all Illumina and Nanopore reads concatenated from the seven viromes (assembly 10), a baseline with the long reads for assembly and polished with all short reads (assembly 11), and a baseline with the long reads for assembly and polished with short reads from only the baseline (assembly 12). Assemblies 1–11 are listed in Figure 1 with their respective assemblers, and assemblies 1–12 are listed in Appendix A.

The distributions of putative viral and nonviral contigs and bases were identified with VirSorter2 in all assemblies (Figure 1A,B and Table 1; detailed data for individual assemblies in Appendix A). This showed that >75% of the assembled contigs were identified as viral in each assembly (Figure 1A and Appendix A), and, of the total 173,915 contigs assembled, 58% or 101,068 were identified as putative viral (Table 1). The viral contigs consisted of relatively long contigs as >84% of the bases were identified as viral in each assembly (Figure 1B and Appendix A). The baseline Canu (assembly 11) had the fewest total contigs (2297) but was the assembly with the least contamination of nonviral contigs (113) and bases (1.1% nonviral Mbp vs. 98.9% viral Mbp) (Appendix A). These results highlight the importance of long reads, corrected with short reads, as the baseline metaSPAdes (assembly 1) using only Illumina reads resulted in a total of 15,782 contigs, where 75.2% were putative viral and 24.8% of the Mbp were putative nonviral (Appendix A). The amount of putative viral vs. nonviral Mbp correlated with the length of the contigs for the two baseline assemblies as N50 was 2901 bp and 21,781 bp for metaSPAdes (assembly 1) and Canu (assembly 11), respectively. Appendix A show that there was little to no change in the identified putative viral contigs and Mbp by VirSorter2 between assemblies 11 and 12. Assembly 12 was added to compare only baseline Nanopore reads vs. baseline Illumina reads to ensure that the longer contigs and lesser nonviral Mbp were a result of the long reads and not of polishing with all short reads from the seven viromes. Only assemblies 1–11 were used for further analysis.

Each of the 11 assemblies was clustered into viral populations as viral operational taxonomical units (vOTUs) with 95% identity over 80% of the genome, presumably at the species level (Figure 2A and Appendix A). Similarly, the total 101,068 VirSorter2 putative viral contigs from the 11 assemblies (Table 1) were clustered into 35,070 vOTUs (Figure 2B and Appendix A). The representative vOTUs of each viral population were put through CheckV (v0.6.0) to assess the contig quality in terms of phage genome completeness. Each sample’s vOTUs were classified as either putative high-quality, medium-quality, low-quality, or not determined (Figure 2).

The Canu assembler using the long reads and polished with the short reads gave a much greater number of high-quality and medium-quality vOTUs than the assemblies using short reads only (assemblies 1–10; Figure 2A). The baseline Canu assembly (assembly 11) generated 2004 vOTUs, while the baseline metaSPAdes (assembly 1) generated 14,788, which is close to seven times more (Appendix A). Of the 2004 vOTUs from the baseline Canu assembly, 116 were denoted as high-quality and 171 were denoted as medium-quality. Contrastingly, the baseline metaSPAdes (assembly 1) only had 29 high-quality and 51 medium-quality vOTUs despite the total contig number being seven times higher (Appendix A). Only by using all concatenated Illumina reads and scaffolding with Nanopore reads (assembly 10) were more high-quality and medium-quality vOTUs observed (132 and 196, respectively), albeit out of a total of 36,480 contigs (~18 times higher contig number; see Appendix A). This means that the hybrid metaSPAdes (assembly 10) with Illumina and Nanopore reads from these seven viromes gave 0.36% high-quality vOTUs, while the Canu reads (assembly 11) resulted in 5.8%. This shows the difference in assembling with short reads and scaffolding with long reads vs. assembling with long reads and polish with short reads (assembly 10 vs. assembly 11).

When running CheckV on the total 35,070 vOTUs from this study, we found that the percentage of putative high-quality and medium-quality contigs was 0.64% and 0.97%, respectively (Figure 2B). These percentages would, with high certainty, have been much higher if Nanopore sequencing was possible for all seven viromes. Our results highlight (not surprisingly) how long reads raise the quality in terms of assembling phage genome completeness in the baseline virome (assemblies 1, 10, and 11). By using only or mostly short reads, many of the complete/high-quality viral contigs were possibly fragmented and predicted as low-quality.

According to the Kraken v2.1.1 results, we observed reads taxonomically classified as bacterial, especially in the DC3000 + A, B, C viromes, potentially reflected in the difference in DNA concentration of said samples (Appendix A). The bacterial genus with the most classified reads throughout all viromes was *Lactobacillus.* Furthermore, the three DC3000 + samples also had reads mapping to the DC3000 strain. Filtering steps to remove bacterial reads before assembly in all seven viromes are explained in Section 2.5. Nanopore sequencing was feasible on the baseline OW virome, and 12 Gb of data in total were generated from the flow cell. The other six viromes did not extract enough DNA for Nanopore sequencing (Appendix A).

### 3.2. Differences in Viral Composition Correlates with the Presence/Absence of vOTUs Rather Than High/Low Abundance

To remove false-positive viral contigs post-VirSorter, all representative viral contigs were mapped against the NCBI RefSeq bacterial genomes database. All contigs covered >40% by a bacterial RefSeq hit were removed. This ensured all contigs assigned to *Pseudomonas*, *Lactobacillus,* or other bacterial species would be considered putative bacterial contamination and were not included for further analysis. All representative viral contigs <5 kbp (unless they were denoted high-quality or medium-quality by CheckV) were removed for further analysis to avoid noise from small fragments. This resulted in a total of 3515 vOTUs. Quality-controlled Illumina reads from the baseline, and each of the DC3000 + and DC3000 − viromes was down sampled to the same number of reads. Reads from each of these seven samples were then mapped against the 3515 filtered vOTUs to calculate the mean depth.

For each of the seven viromes, we observed 1387 to 1547 vOTUs (Appendix A). To detect how similar their compositions were, we performed a hierarchical clustering based on the Bray–Curtis dissimilarities between each sample (Figure 3). We observed a difference in viral population composition between the DC3000 + and − samples and the baseline. However, the DC3000 − had one sample (DC3000 − A) that clustered closer together with the DC3000 + viromes in regard to composition (Figure 3A–C). Changing beta-diversity to capture differences due to presence/absence (Jaccard) revealed similar results both in terms of hierarchical clustering and in relation to experimental design [49]. When comparing Jaccard and Bray–Curtis distance matrices with a mantel test, we found a strong positive correlation (*r* = 0.93, *p*-value = 0.001). This indicates that the virome differences are driven by the presence/absence of certain viruses and, to a lesser extent, their abundance (Appendix A). Thus, DC3000 − A could cluster together with the DC3000 + samples due to the stochasticity in which viral populations can evolve [50,51]. No bacterial host was added to this sample, whereas only clean media and sterile water were used for the virome. Nevertheless, the two viromes DC3000 − B and C had a more similar viral composition to the baseline virome than any of the DC3000 + viromes (Figure 3A,B), and the overall difference between sampling groups was trending but not significant (ANOSIM test statistic = 0.58, *p*-value = 0.085, Figure 3D). Hence, the overall viral population composition was altered due to the addition of DC3000 to the sample but was also affected by the incubation in growth media, as the difference was not significant between the virome sampling groups. The possibility for a nonbacterial free OW virome caused by the filtering step is discussed in Section 4, as it can be a potential factor in how the overall composition of the viral population is altered.

### 3.3. Most Abundant vOTUs Indicate a Population Change

The cumulative abundance indicated that the 10 most abundant contigs constituted ~10% of the reads in the baseline virome, while they constituted ~12.5% in the DC3000 + and DC3000 − viromes. Hence, the 10 most abundant vOTUs in the DC3000 + and DC3000 − accounted for around 2.5% more reads than the 10 most abundant contigs in the baseline assembly (Figure 4). A joined cumulative abundance graph of all contigs from the DC3000 +, DC3000 −, and baseline metaSPAdes assemblies is shown in Appendix A.

The two contigs NODE_532 and NODE_5 were highly abundant in DC3000 + (1.04% and 0.87% relative abundance, respectively; Figure 4B–D). NODE_532 had lower read abundance in the baseline virome and DC3000 − (0.41% and 0.77%, respectively), and NODE_5 was completely absent in both samples (0.00% and 0.00%, respectively) (Figure 4A). A nucleotide blast alignment with the 99,050 bp long NODE_5 showed that it has a query coverage and percentage identity of 89% and 94.19%, respectively, as well as an e-value of 0, to the *Pseudomonas* phage VMC (Accession no. LN887844.1). NODE_5 was also observed to cluster together with one (DrKristoffer) of the five plaque isolated phages, which is further elaborated in Section 3.4. NODE_532 was not absent in the baseline virome or DC3000 − (0.41% and 0.77% relative abundance, respectively) compared to NODE_5 (0.00% and 0.00% relative abundance, respectively) (Figure 4A). A blast alignment showed that it aligned to the *Lactobacillus* phage 3-521 with a query coverage and percentage identity of 4% and 74.72%, respectively, as well as an e-value of 4 × 10^−41^ (Accession no. NC_048753.1). As the query coverage was only 4%, it is likely that only one gene is shared between NODE_532 and *Lactobacillus* phage 3-521.

In the group of 10 most abundant contigs for each of the viromes, three (tig00000010, tig00000787, and NODE_118) were highly abundant in all three viromes, while five (tig00000595, NODE_477, tig00000357, tig00000497, and tig00000860) were highly abundant in DC3000 + and DC3000 − (Figure 4B–D). The baseline had seven contigs that were highly abundant but not within the group of 10 most abundant contigs in the DC3000 + and DC3000 − viromes. Two contigs, tig00000010 and tig00000787, were in the top four contigs in all three viromes, but they shifted as the tig00000010 relative abundance was reduced to 0.37% in both DC3000 + and DC3000 −, whereas tig00000787 had a 1.4% higher depth coverage in both viromes compared to the baseline. The tig0000787 contig was also 0.85% and 0.75% higher in its accumulative abundance compared to the second most abundant contig, in DC3000 + and DC3000 −, respectively. These results indicate an abundance change potentially due to the LB medium incubation, as the shift was similar in both samples (Figure 4B–D).

### 3.4. DrKristoffer, Hovsa, and OtownIsak Cluster Together with Two Contigs from the OW Viromes

In parallel with the viromes, we isolated five single *P. syringae* phages from an aliquot of the OW sample, also using the DC3000 strain as a host. These were included in the analysis to observe whether the isolated phages dominated the enriched population when the virome was challenged with the DC3000 strain. Their taxonomy at the family and genus level is denoted in Table 2, together with their genome size, cluster representative, and blast comparison. All reads (not subsampled) were mapped to the five genomes and are listed in Appendix A.

Of the five phages that made plaques on DC3000, only SummerBoyErik was incubated overnight with DC3000 in liquid media prior to plaque assay. GhostToast, Hovsa, and OtownIsak were all isolated with direct plating, whereas, for the isolation of DrKristoffer, the OW sample was concentrated using PEG precipitation before plaque assay.

The genomes of the five phages were added to the analysis to observe which contigs they clustered together with (presumably at the species level). NODE_5 (99,050 bp), mentioned in Section 3.1, was one of the 10 most abundant contigs in DC3000 +. NODE_5 was representative of the vOTU cluster containing DrKristoffer, and a blast alignment between NODE_5 and DrKristoffer showed a coverage and percentage identity of 91% and 93.61%, respectively, as well as an e-value of 0 (Table 2). Hence, NODE_5 is potentially a phage from the same species as DrKristoffer. GhostToast and SummerBoyErik did not cluster to any putative viral contig in the viromes or to each other. The two phages Hovsa and OtownIsak were members of the cluster represented by NODE_16 (40,386 bp). Due to the coverage and percentage identity of 100%, the phage OtownIsak and NODE_16 could be considered the same phage (Table 2). However, Hovsa had a coverage and percent identity with NODE_16 of 94% and 97.41%, respectively (Table 2). These high values for both phages against NODE_16 suggest that they belong to the same species as NODE_16, but it is not possible to discern whether NODE_16 is OtownIsak or Hovsa. GhostToast and SummerBoyErik are phages of the same family (*Autographiviridae*) as NODE_16/OtownIsak/Hovsa, but they are not related at the species level to any contig in the viromes as DrKristoffer and NODE_16/OtownIsak/Hovsa. Mapping the total reads from each virome sample against each phage genome showed that DrKristoffer/NODE_5 was only present in DC3000 + A, but NODE_16/OtownIsak/Hovsa was present in all seven viromes (Appendix A). When comparing the isolated member of NODE_16 (OtownIsak and Hovsa), OtownIsak had the most reads mapping its genome. This is, however, consistent with its ~100% percentage identity to NODE_16 and how BBmap classifies ambiguous reads according to the best scoring sites (Table 2 and Appendix A) [52]. This could indicate that, even though we isolated these phages from the same OW viromes, three of them (SummerBoyErik, GhostToast, and DrKristoffer/NODE_5) were not highly abundant after incubation with DC3000 in all three replicates or the baseline.

### 3.5. Pseudomonas Phages Are Significantly Enriched When Exposed to DC3000

From the 3515 vOTUs (from Section 3.2), 2063 were present in at least one of the samples, considering that at least 80% of the vOTU length needs to be covered to be considered present. Hence, 2063 vOTUs were included in the abundance comparison. Their *p*-values were plotted against the log_2_ fold change (Log_2_FC), where each dot represents contigs becoming either more absent (Log_2_FC < 0) or more present (Log_2_FC > 0) when the virome was incubated with the DC3000 host (Figure 5A).

A total of eight contigs had a nominal significant *p*-value (α = 0.05) with a Log_2_FC above 1 (green dots, Figure 5A). These were the vOTUs enriched with the presence of the DC3000 strain. On the other hand, 11 contigs had a significant *p*-value with a Log_2_FC below −1, which were all phages reduced in abundance when the host was added (red dots, Figure 5A). The most abundant contig with a significant Log_2_FC of 1.68 was NODE_16, which we identified as the same species as OtownIsak and Hovsa in Section 3.3 (Figure 5A and Table 2). It was the most significant contig (*p* < 2 × 10^−7^) and had a high read abundance in all three viromes of DC3000 + (Figure 5B). NODE_5 had a higher Log_2_FC of 1.95 but was not significant (*p*-value = 0.17). This was caused by its high abundance in DC3000 + A, which it did not have in B and C; therefore, it was not significantly enriched compared to the DC3000 − viromes (Figure 5B). NODE_5 clustered together with DrKristoffer as they are the same species with a coverage of 91% (Figure 5A and Table 2). The last two plaque-isolated phages, GhostToast and SummerBoyErik, had a very low abundance through all triplicates of DC3000 + and DC3000 −, compared to the three other single isolated phages (Figure 5B and Appendix A). The seven additional contigs with a significant abundance and Log_2_FC >1 in Figure 5 are all listed in Appendix A, together with contigs with a significant abundance and Log_2_FC <−1. Contigs that had a significant abundance but a Log_2_FC between −1 and 1 are listed in Appendix A.

Using CRISPR spacer sequence analysis, we could assign hosts at the species level for a total of 503 out of the 2063 vOTUs. Of the 503, 32 (6.36%) were putatively assigned as *Pseudomonas* phages and 209 (41.55%) were putatively assigned as *Lactobacillus* phages. Moreover, the vOTUs were blasted against a database consisting of 510 *Pseudomonas* phages deposited in NCBI to further distinguish *Pseudomonas* phages. A total of 87 vOTUs were covered more than 60% by a contig from NCBI. The 32 CRISPR spacer matching sequences and the 87 NCBI putative *Pseudomonas* phages (119 total) were manually curated as explained in Section 2.9. Of the previous 119 vOTUs, 47 could be classified with high confidence as *Pseudomonas* phages, denoted as yellow dots (Figure 5A and Appendix A). Moreover, 36 of the 47 (77%) curated *Pseudomonas* phages (yellow dots, Figure 5A) and 92 of the 209 (44%) CRISPR spacer matching *Lactobacillus* phages (blue dots, Figure 5A) had a Log_2_FC >0. We used Fisher’s exact test to determine whether the proportion of enriched *Pseudomonas* phages was significant, revealing an odds ratio of 2.39 (Fisher’s exact test CI: 1.16–5.30, *p* = 0.015), indicating a significant enrichment of *Pseudomonas* phages when incubated with a host organism, compared to *Lactobacillus* as a phage–host reference.

## 4. Discussion

When isolating single phages, the most used method is the plaque assay [21,30]. The plaque assay is also used to quantify recovered phages after virome extraction when the virome sample is spiked with a known phage in a high concentration to assess how efficient the virome extraction is [53]. To our knowledge, it is not well known whether plaque-isolated phages are the most responsive to the host of interest or if they are simply the ones most capable of lysing the bacterial host in solid media.

Host–virus dynamics in liquid media have previously been analyzed with dilute-to-extinction isolation, but we hypothesized that, when a virome is spiked in liquid media with a bacterial strain, novel to the phage community, there should be a shift in abundance toward phages targeting that strain [19,54]. Furthermore, we also hypothesized that the most enriched phages in the spiked virome are not necessarily those observed by plaque assay isolation. In this study, we mixed an overnight culture of *P. syringae* DC3000 with a filtered OW sample containing the virome in liquid LB medium at room temperature for 8 h. We added the same OW virome sample to liquid LB medium without an overnight culture as a control. By comparing these two types of viromes, we observed which phages are enriched by the *P. syringae* strain. Subsequently, we isolated five phages using the traditional plaque assay from the same virome–host matrix. This allowed us to compare them to the most enriched vOTUs in the spiked virome. If they were highly abundant in the spiked virome, it indicated that they are very effective in infection of the host in both types of media (solid and liquid). However, if they were in low abundance, it indicated out-competition of these when mixed with other phages in liquid media and, therefore, much lower infection effectiveness when incubated with the host in solid media. Studying the shift of viromes when exposed to a specific bacterial host and analyzing how effective the single isolated phages are in a different type of medium will increase our general knowledge about phages and improve the methods we use to isolate phages that are nonplaque-forming.

The changes in the two types of viromes (DC3000 + and DC3000 −) in this study were examined at a fixed time point corresponding to a few initial infection cycles in the DC3000 + samples. The incubation time of 8 h allowed the DC3000 population to have doubled approximately twice, as its generation time was shown to range from 2–6 h from inoculation [26]. We expect this incubation time to be long enough to include phages with long latent periods and sufficiently short to prohibit a scenario of a single phage dominating the virome completely. Another interesting aspect would be to study the dynamic of changes in a time series. Previous single-phage studies have shown a correlation between host density and the quality and optimality of phage growth over time [55,56]. Hence, DC3000 + and DC3000 − viromes from different time points might highlight distinct dynamics in the virome composition.

We assembled a large virome from OW with 2063 putative vOTUs represented in at least one of the triplicates with a read coverage of >80% that could be used for read abundance analysis. From this analysis, we observed eight putative phages that were significantly upregulated with a Log_2_FC >1 and 11 that were significantly downregulated with a Log_2_FC <−1 (Figure 5A). Five dominant plaque-forming phages could be isolated from the OW sample using DC3000 as host, and the phages OtownIsak and/or Hovsa were indeed identified in the spiked virome as one of the most enriched vOTUs to DC3000 (Table 2, Figure 5A, and Appendix A). The phage DrKristoffer had a high Log2FC but was only highly abundant in one of the triplicates of DC3000 +, whereas the last two phages (SummerboyErik and GhostToast) had low abundance in all triplicates of both DC3000 − and + and did not have a breadth coverage above 80% (Appendix A). This suggests that it is not necessarily the most abundant phages that are isolated by the plaque assay. We show that comparative viromics can indicate significant host-specific dynamics, as we observed that the proportion of enriched *Pseudomonas* phages was significant despite a large background of mainly *Lactobacillus* phages (Figure 5A).

A total of 3515 putative vOTUs were identified across all seven viromes (Appendix A shows the number of vOTUs in the individual samples). When comparing the vOTU composition in the individual viromes, we found a strong positive correlation between the Bray–Curtis and Jaccard distance matrices (*r* = 0.93, *p*-value = 0.001), suggesting that the clustering is due to the presence/absence of vOTUs between the samples rather than abundance correlated. We observed that the composition of DC3000 − A was very similar to the DC3000 + triplicates (Figure 3). As mentioned previously, we filtered the samples with 0.45 μm PVDF filters to deplete bacteria, whereby this filter size is not small enough to remove all bacteria, yet large enough to include larger phages otherwise lost with a pore size of 0.22 μm [57,58,59,60,61]. If the bacterial contamination was similar to the *Pseudomonas* DC3000 in any way, the sample could potentially evolve toward a similar composition as the DC3000 + viromes [57]. As the DC3000 − B and C more closely resembled the baseline virome, we assumed that it was due to a stochastic variable, such as random bacteria passing through the filter, affecting the DC3000 − A (Figure 3A). The choice of a larger pore size could potentially also be related to the 10–11% reads mapping to *Lactobacillus* in all seven viromes, as we saw that DC3000 + and DC3000 − had a more similar shift in vOTU composition compared to the baseline (Figure 3C). If the OW sample was completely depleted of all bacteria via filtration, it would be more likely that the DC3000 − triplicates would be closer to the baseline in vOTU composition.

A total of 209 *Lactobacillus* phages were identified with CRISPR host prediction in the 2063 vOTUs, where the second most present genus was *Pseudomonas* phages with 47 (Appendix A). The OW sample has a low pH value (≤3 pH), and, as *Lactobacilli* prefer a low pH value and are often associated with plant fermentation, it likely is one of the dominant genera in the bacterial community [62,63]. We used Fisher’s exact test to determine if the proportion of enriched *Pseudomonas* phages was significant in the DC3000 + viromes. This showed that, even though those 36 *Pseudomonas* phages did not significantly change in abundance, their proportion that increased was significantly higher than that of those that decreased. On average, there were 2.39 more putative *Pseudomonas* phages with increased abundance in the DC3000 + viromes (95% CI: 1.16–5.30). Hence, we showed a significant host-specific shift in the OW viromes when incubated in a liquid medium together with the DC3000 strain.

A total of 14 vOTUs had a significant difference in read abundance but a Log_2_FC between 1 and −1, of which none aligned to *Pseudomonas* phages, but six were similar to *Lactobacillus* phages (Appendix A). Additionally, 11 contigs were significantly different in their read abundance with a Log_2_FC <−1, of which four were similar to *Lactobacillus* phages and none were similar to *Pseudomonas* phages (Appendix A). One of those four contigs was the complete genome of *Lactobacillus* phage 3-521 (accession no. NC_048753.1) isolated from Irish wastewater [64].

Eight contigs had a highly significant *p*-value (α = 0.05) with a Log_2_FC above 1 (Appendix A). Three of those were putative *Lactobacillus* phages, while one (NODE_3) was the complete genome of the *Enterobacteria* phage CAjan (accession no. KP064094.1) isolated from rat feces [65]. The most significantly abundant of those eight was NODE_16, potentially the complete genome of the phage OtownIsak and/or Hovsa (Figure 5A). Both phages were isolated using the plaque assay as described earlier, together with GhostToast, SummerboyErik, and DrKristoffer. The phage DrKristoffer is the same species as NODE_5, which had a higher Log2FC than NODE_16 but an insignificant *p*-value due to it having a high read abundance in only one of the DC3000 + viromes. Both GhostToast and SummerboyErik had a low abundance in all triplicates of DC3000 + and DC3000 −. If all five phages were the most effective at infecting and lysing the DC3000 strain in liquid media, their abundance would have been significantly high in all DC3000 + triplicates. As only OtownIsak or Hovsa were found to have a significant read abundance, it is plausible to assume that the three other phages are not effective in lysing in liquid media but are able to lyse in solid media as they can form plaques. Surprisingly, SummerboyErik was isolated by enrichment in liquid LB before direct plating with soft agar double overlay. From this, we would have expected SummerboyErik to be highly abundant after 8 h, but perhaps the phage has a longer lag phase and would have been significantly abundant in the DC3000 + viromes if left for longer. Another reason could be a limited burst size that would mask the abundance of SummerBoyErik to highly abundant phages.

By extracting the virome directly from the OW sample as the baseline virome, we were able to acquire long reads that included, in theory, all the initial phages present in the OW sample. The DNA concentration from the DC3000 + and DC3000 − triplicates were not sufficient for Nanopore sequencing (Appendix A). Multiple displacement amplification (MDA) was not an option for this study as it has been shown to introduce bias such as over-amplification of small circular genomes and under-amplification of DNA with high GC content in viromes from humans saliva [66]. The comparison of virome quality from the metaSPAdes assemblies and the Canu assembly further supports the importance of long-read data when performing metaviromics projects (Figure 1 and Figure 4). Even though the baseline metaSPAdes (assembly 1) has more vOTUs (14,788) than the baseline Canu (assembly 11 with 2004), the N50 (metaSPAdes 2343 bp and Canu 21,775 bp) and the number of high-quality vOTUs (metaSPAdes 29 and Canu 116) are still greater for the Canu assembly. Furthermore, 95.7% of the length of the Canu assembled contigs were covered by Illumina reads, showing that the information obtained from both platforms was similar. Thus, this difference was possibly due to the fragmentation because of the presence of microdiversity on the viral populations that cannot be resolved with only short reads [67]. When adding the long reads, we saw an increase from 0.18% to 5.79% of high-quality phage genomes compared to using only short reads despite the higher difference in the read amount (Appendix A, Figure 2, and Appendix A). For studies not doing comparative analysis as in our study, amplification is possible. Oxford Nanopore Technologies offers library kits with lower-input DNA requirements (Rapid PCR Barcoding Kit SQK-RPB004 used in this study), but it is still not close to what is possible with the low-input DNA Illumina library preparations (Nextera^®^ XT DNA kit) [25,68]. Viromes sequenced with a combination of long and short reads will elevate the quality of UViGs in the public databases compared to only using short reeds (Figure 2) [69]. Better databases with high-quality phage genomes would also raise the quality of the available bioinformatics tools for phage identification in metavirome data. We attempted to assemble the seven viromes before removing the bacterial contamination to investigate how VirSorter2 and CheckV would assess the assembled contigs and if the former would be able to remove the bacterial contigs [41,42]. Both tools identified contigs that, when blasted, had 100% alignment to bacterial genomes and none to phage genomes. Hence, there was harsh removal of all putative bacterial reads, although we lost potential data from lysogenic phages that we observed before the filtering.

## 5. Conclusions

Our results show that the DC3000 + and DC3000 − viromes reveal significant host-specific dynamics of the phages when incubated for 8 h in liquid media. We also showed that three out of five phages isolated individually from the same sample virome using the plaque assay were ineffective in terms of lysis in liquid media or outcompeted by phages from the same virome. This has to be considered when deciding if single isolated phages are good potential candidates for treatment in phage therapy or biocontrol [17,29,70,71,72]. Furthermore, our analysis highlighted how much long-read sequencing data improve the assembly of viral metagenomes, as we saw an increase of 5.64% in the proportion of complete phage genomes from long-read assembly compared to short-read-only assemblies (*p*-value < 0.0001, Figure 2).

## Figures and Tables

**Figure 1 viruses-13-00959-f001:**
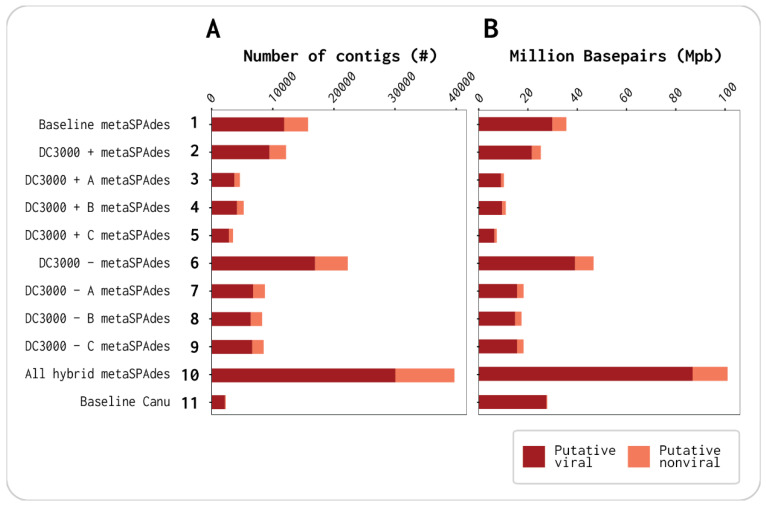
Identification of putative viral and putative nonviral contig count and million base pairs (Mbp) in each of the 11 assemblies by VirSorter2: (**A**) distribution of the number of putative viral vs. nonviral contigs; (**B**) distribution of putative viral vs. nonviral Mbp.

**Figure 2 viruses-13-00959-f002:**
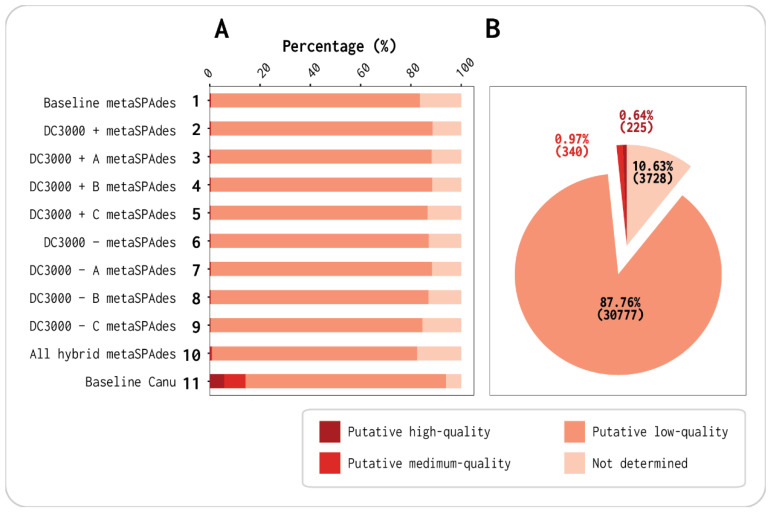
Quality identification by viral contigs with CheckV: (**A**) percentage of quality distribution of viral contigs in each of the 11 assemblies; (**B**) percentage of quality distribution of viral contigs for joined assemblies.

**Figure 3 viruses-13-00959-f003:**
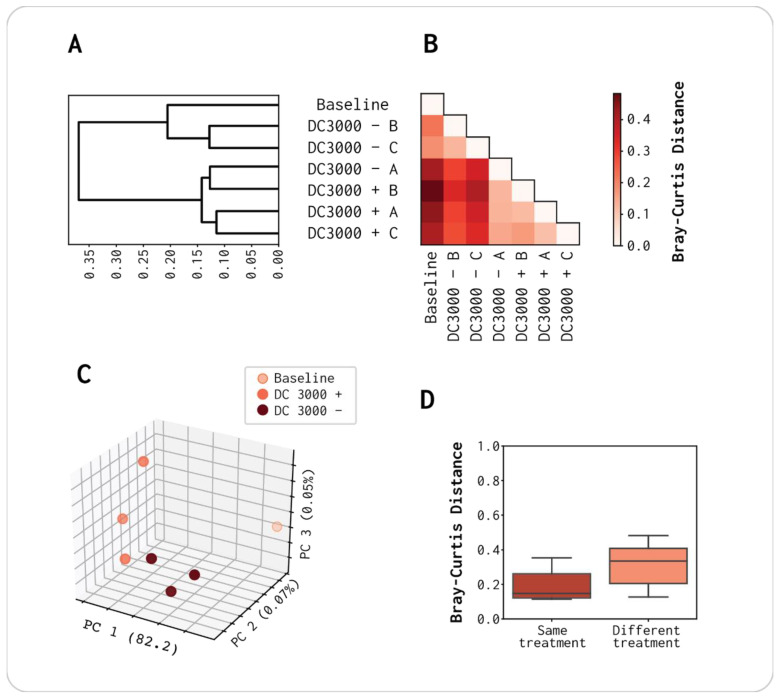
Viral OTU composition of the population present in the three replicates of virome DC3000 +, DC3000 −, and the baseline virome: (**A**) hierarchical clustering of vOTUs based on Bray–Curtis distances; (**B**) pairwise Bray–Curtis distances; (**C**) PCoA plot; (**D**) comparison of treatment incubation of OW in LB media at room temperature for 8 h with or without an overnight culture of DC3000. ANOSIM test statistic = 0.58 (*p*-value = 0.085).

**Figure 4 viruses-13-00959-f004:**
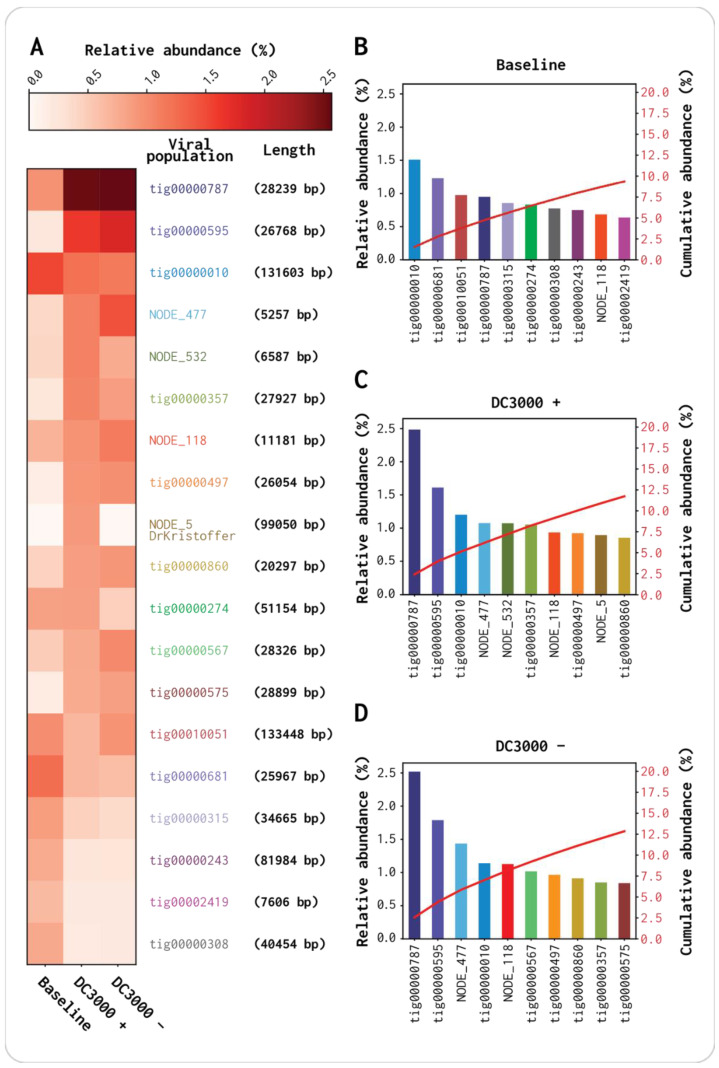
Abundance comparison of 10 most abundant contigs for the three viromes (baseline, DC3000 +, and DC3000 −). All contigs were putative viral, as identified by VirSorter2, and passed through filtering; they had <40% query coverage to the bacterial database and were >5 kbp: (**A**) heatmap of read-abundance correspondent to the 10 contigs from each of the three viromes; (**B**–**D**) relative abundance percentage (left *Y*-axis) and cumulative abundance percentage (right *Y*-axis) of the 10 most abundant contigs in the baseline virome; (**B**) baseline virome; (**C**) DC3000 + virome; (**D**) DC3000 − virome. It has to be noted that we did not account for whether these contigs could come from the same viral genome as fragments from the assembly.

**Figure 5 viruses-13-00959-f005:**
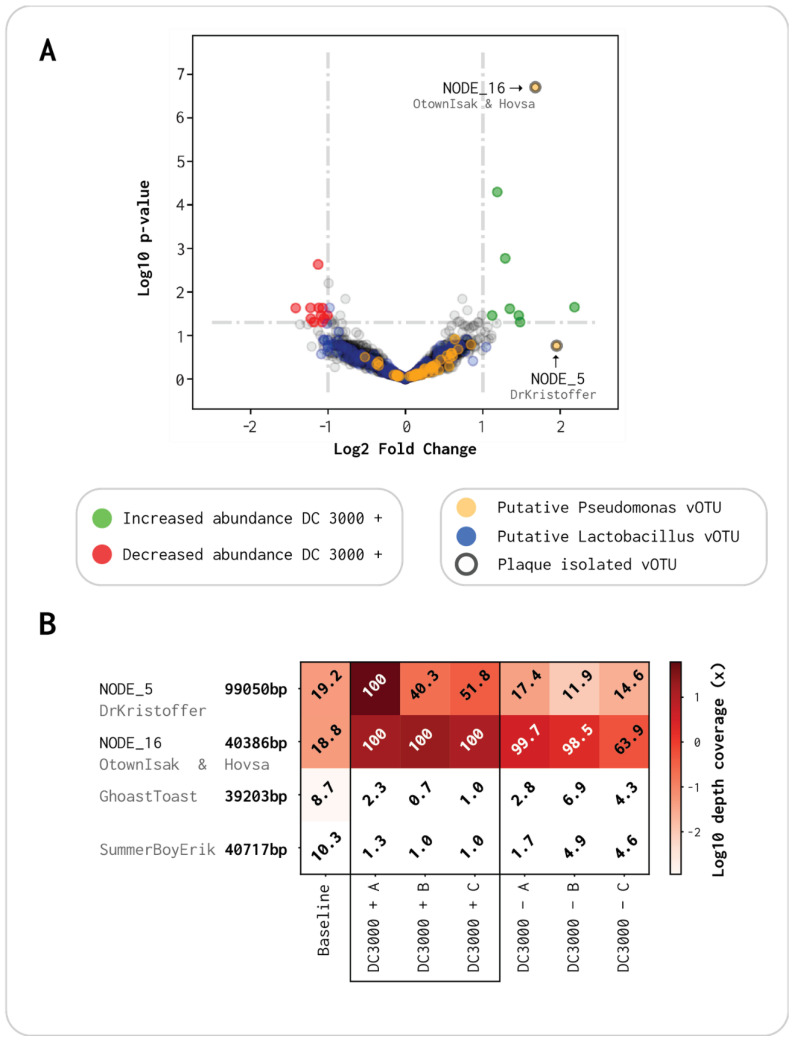
Volcano plot of abundance comparison between DC3000 + and DC3000 −: (**A**) the *X*-axis is the log_2_ fold change (Log_2_FC) in read abundance between the two viromes, and the *Y*-axis is the log_10_ *p*-value for read abundance in the triplicates. **Ringed dots:** plaque-forming phages; **yellow dots**: putative *Pseudomonas* phages; **blue dots:** putative *Lactobacillus* phages; **green dots**: viral contigs that increased by >1 Log_2_FC with a significant *p*-value; **red dots:** viral contigs that were reduced by <−1 Log_2_FC with a significant *p*-value; (**B**) read abundance for the five plaque-forming phages throughout the seven viromes. White numbers indicate contigs with a breadth coverage ≥80%, which were presumed present, while black values indicate contigs with <80% breadth coverage, regarded as not present.

**Table 1 viruses-13-00959-t001:** Data of total contigs and total Mbp from the merged 11 assemblies. Putative viral and putative nonviral contigs and Mbp were identified by VirSorter2 and are visualized for each assembly in Figure 1. Data for the contig and Mbp distributions of individual assemblies are presented in Appendix A. Assembly 12 is excluded.

	No. of Contigs	%		No. of bp	%
Total	131,073	100	Total	317,732,011	100
Putative viral	101,068	77.1	Putative viral	275,137,398	86.6
Putative nonviral	30,005	22.9	Putative nonviral	42,594,613	13.4

**Table 2 viruses-13-00959-t002:** Overview of the five plaque-forming phages isolated against DC3000 from the same OW sample used for DC3000 + and DC3000 − viromes. All blast hits had an e-value of ~0. GhostToast and SummerBoyErik did not have any cluster representatives.

Phage	Genome Size	Family/Genus	Cluster Representative (CR)	Coverage of CR	Percent Identity to CR
DrKristoffer	98,997	*Myoviridae/Otagovirus*	NODE_5	91%	93.61%
Hovsa	40,290	*Autographiviridae/Ghunavirus*	NODE_16	94%	97.41%
OtownIsak	40,331	*Autographiviridae/Ghunavirus*	NODE_16	100%	99.99%
GhostToast	39,235	*Autographiviridae/Pifdecavirus*	-	-	-
SummerBoyErik	40,716	*Autographiviridae/Pifdecavirus*	-	-	-

## Data Availability

All reads used in this study are deposited at SRA and published under the BioProject ID PRJNA724013.

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
