# Peer review of "Metaviromes Reveal the Dynamics of Pseudomonas Host-Specific Phages Cultured and Uncultured by Plaque Assay"

_viruses, 2021, doi:10.3390/v13060959_

Round 1

Reviewer 1 Report

Overall the manuscript provides a comprehensive study comparing the success rate of isolating phages via plaque assays to a metagenomics approach in evaluating phage abundance in a sample (with and without potential host) and possible implications in phage therapy and biocontrol. The approach taken utilizes metaviromics while acknowledging the importance of traditional plaque assay to isolate phages for further use. One detail that might be of interest is how the phage dynamic changes over time, rather than using a fixed time point for analysis.

Author Response

Dear Reviewer #1

We cherish that you took the time to read our manuscript and highly appreciate your comments. 

Changes have been made throughout the manuscript to improve the English language and style. These changes are in lines such as 18-19, 24, 30, 101, 113, 131, 156, 222-223, 255, and 273. A full response letter covering all changes made has been attached. Please see the attachment. 

We agree with you that it would be interesting to investigating the phage dynamic over time, and not just one fixed time point. This fixed time point in our study corresponds to a few initial rounds of infection and burst. The dynamic changes over time would be for a future similar study, that would indicate if the phage dynamics of the virome shifts multiple times when subjected to a host. We have added a small paragraph in the discussion from line 608-619. 

Kind regards

The Authors

Reviewer 2 Report

This was a nicely presented paper. Easy to follow the workflow and an interesting read. There are some minor English issues eg line46 should read as 'correspond'; line523 should read as 'significantly'. Such errors occur thoughout the manuscript.

Author Response

Dear Reviewer #2

We cherish that you took the time to read our manuscript and highly appreciate your comments. 

Changes have been made throughout the manuscript to improve the English language and style. These changes are in lines such as 18-19, 24, 30, 101, 113, 131, 156, 222-223, 255, and 273. Including those in pointed out in the comments. A full response letter covering all changes made has been attached. Please see the attachment. 

Kind regards

The Authors
